# Interface-mediated Kirkendall effect and nanoscale void migration in bimetallic nanoparticles during interdiffusion

See Wee Chee [1,2], Zicong Marvin Wong [3], Zhaslan Baraissov[1,2], Shu Fen Tan[1,2], Teck Leong Tan [4] & Utkur Mirsaidov [1,2,5,6]

At elevated temperatures, bimetallic nanomaterials change their morphologies because of the interdiffusion of atomic species, which also alters their properties. The Kirkendall effect (KE) is a well-known phenomenon associated with such interdiffusion. Here, we show how KE can manifest in bimetallic nanoparticles (NPs) by following core–shell NPs of Au and Pd during heat treatment with in situ transmission electron microscopy. Unlike monometallic NPs, these core–shell NPs did not evolve into hollow core NPs. Instead, nanoscale voids formed at the bimetallic interface and then, migrated to the NP surface. Our results show that: (1) the direction of vacancy flow during interdiffusion reverses due to the higher vacancy formation energy of Pd compared to Au, and (2) nanoscale voids migrate during heating, contrary to conventional assumptions of immobile voids and void shrinkage through vacancy emission. Our results illustrate how void behavior in bimetallic NPs can differ from an idealized picture based on atomic fluxes and have important implications for the design of these materials for high-temperature applications.

[1] Department of Physics, National University of Singapore, Singapore 117551, Singapore. [2] Centre for BioImaging Sciences, Department of Biological Sciences, National University of Singapore, Singapore 117557, Singapore. [3] Department of Chemistry, National University of Singapore, Singapore 117543, Singapore. [4] Institute of High Performance Computing, Agency for Science, Technology and Research, Singapore 138632, Singapore. [5] Centre for Advanced 2D Materials and Graphene Research Centre, National University of Singapore, Singapore 117546, Singapore. [6] Department of Materials Science and Engineering, National University of Singapore, Singapore 117575, Singapore. Correspondence and requests for materials should be addressed to U.M. (email: mirsaidov@nus.edu.sg)

Bimetallic nanoparticles (NPs) are an important class of materials for catalysis, plasmonics, and biomedical applications[1]. The morphology and composition of these NPs define their properties, and both features readily change at elevated temperatures due to atomic interdiffusion. For example, the intermixing of different atomic species can lead to strain effects that influence the catalytic performance of NPs[2]. Yet, we still do not have a complete picture of the diffusional processes that occur in NPs[3]. One reason is that these NPs are usually only characterized before and after heating, and so, the transient transformations that take place during heating remain unclear. Insights into these processes are needed for the design of thermally stable nanostructures[4]. The Kirkendall effect[5] (KE) is one process that can modify NP structure during interdiffusion. It is a classic phenomenon in bulk metallurgy that is also associated with void formation[6]. At the nanoscale, it is believed that KE is responsible for the formation of hollow nanostructures during the oxidation of monometallic NPs[7,8] and during NP galvanic replacement reactions[9,10]. Hollow monometallic NPs created by exploiting nanoscale KE can have catalytic performance superior to solid NPs[11].

KE and void formation are, in fact, distinct processes. KE refers to the movement of the bimetallic interface when two attached metals are heated to temperatures where atomic diffusion occurs[5]. This shift is caused by the metals having different interdiffusion rates. Consequently, there is also an associated flux of vacancies towards the faster diffusing metal[6]. Void formation is the result of these vacancies clustering at structural defects in the metal, such as grain boundaries[12,13]. The discovery of KE is significant because it validated the vacancy model for atomic diffusion. KE was also pivotal in explaining the failure of wire-bonds due to void formation[14]. Thus, KE has been studied extensively in bulk metal alloys and bimetallic thin films to mitigate the undesirable consequences of these voids. However, it is unclear how KE manifests in bimetallic NPs. It was reported that Ag–Au[15] and Ag–Pd[16] NPs heated to ~450 °C form voids, but these studies lack clear evidence of KE because the NPs were only examined after heating. Furthermore, questions remain regarding where voids nucleate in NPs. In the bulk, voids are expected to nucleate at grain boundaries and similar defects[12,13]. Since NPs are usually too small to have grain boundaries, it is not certain what structural features serve as the nucleation sites for voids.

Here, we examine how nanoscale diffusion couples of Au and Pd change their morphologies and compositions during heating with in situ scanning transmission electron microscopy (STEM) and broad beam transmission electron microscopy (TEM). Our diffusion couples are core–shell NPs with cuboid cores and with relatively thick shells of 10 nm or more. We choose bimetallic NPs of Au and Pd because they have promising properties for catalytic applications[17–19]. These NPs are synthesized in three arrangements: Au core–Pd shell (Au–Pd), Pd core–Au shell (Pd–Au), and Au–Pd–Au sandwich structures (see the section "Methods"). These NPs serve as ideal model systems because the cuboid core has flat (100) faces, making it easier to interpret interdiffusion of the two metals. The thicker shells also provide a longer pathway for the diffusion of core atoms. We highlight here that Au and Pd form a solid solution at all compositions[20]. Furthermore, according to earlier thin films studies[21,22], the diffusion of Pd in Au is faster than the diffusion of Au in Pd, which suggest a higher likelihood for hollowing to occur due to KE if Pd is at the core[23,24]. By following the same NPs in real time, we can pinpoint the exact mechanisms for structural transformation. We will show that these NPs, in fact, do not evolve in a manner expected for nanoscale KE. Nanoscale voids form at the bimetallic interface for all three NP geometries. Furthermore, these voids

neither grow into a larger void at the core of the NP nor shrink with extended heat treatment. Instead, the voids migrate to the NP surface.

## Results

**Interdiffusion and void dynamics in Au core–Pd shell NPs.** The changes in morphology of the Au–Pd core–shell NPs during heating are described in Fig. 1. These NPs are not expected to develop internal voids because the faster diffusing Pd is on the outside. In Fig. 1a, we show STEM images and their corresponding elemental maps from energy dispersive X-ray spectroscopy (EDX) for a NP held at 500 °C for accumulated heating durations of 5 and 25 min. Au appears brighter in STEM images because it has a higher $Z$ of 79 versus 46 for Pd. Although there are small voids (darker features) on the Pd side of the as-synthesized samples, heating led to additional voids (highlighted with black dashed boxes in the STEM images) that did not appear to be associated with the original ones. These new voids stick to the Au side of the interface and are, in general, round and no more than a few nanometers in size.

The interdiffusion of Au into Pd in this NP as a function of temperature and heating time is best revealed with EDX. Figure 1b summarizes the diffusion coefficients calculated from several sets of NPs heated to different temperatures (Supplementary Methods). The diffusion coefficients of Au diffusing into Pd was estimated by fitting elemental maps of the same NP at different heating times with a two-sided error function (inset in Fig. 1b, Supplementary Eq. (1)). The maps show the expected outward shift of the bimetallic interface and that the NPs eventually became homogenous alloys with extended heating (Supplementary Fig. 2). From a linear Arrhenius fit of the diffusion coefficients, we obtain the activation energy of $E_a = 1.6 \pm 0.2$ eV for Au diffusion in Pd. For comparison, the activation energies reported for bulk lattice diffusion and grain boundary diffusion in this system are 1.8 and 0.9 eV, respectively[21,22]. More surprisingly, the STEM image sequences from these experiments suggest that the nanoscale voids were mobile during heating and that they moved towards the NP surface, contrary to earlier microscopic studies that indicated the formation of stable voids in bimetallic NPs[15,16].

To reveal details of the void dynamics, we performed in situ TEM experiments where the heating profiles consist of alternating temperature holds and continuous ramps to higher temperatures (an example is presented in Supplementary Fig. 3). The NPs showed the most dynamic changes in morphology when they were heated to 500 °C and above (Fig. 1c) (see Supplementary Fig. 4 for images acquired at lower temperatures). The image sequence in Fig. 1c shows that the core had become indistinct at 600 °C and alternating bands of light and dark contrast indicative of bend contours due to lattice strain appear[25]. Similar to the STEM experiments, we observed the formation of interfacial voids in the Au–Pd NPs. These voids moved when we held the NPs at a high temperature for longer periods or further increased the temperature. This behavior is illustrated in Fig. 1d, e (Supplementary Movie 1), where we raised the temperature to 650 °C. Note that the displacements did not occur continuously over the 400 s time interval; instead, they took place intermittently within a couple of seconds. This void was eventually annihilated at the NP surface. The observed behavior suggests that the void was pinned within the NP and only moved under specific conditions.

The transient migration of pinned voids coincides with the increased dynamics of dislocations that formed during heating[26]. Supplementary Movie 2 and Supplementary Fig. 5a show a Au–Pd NP that exhibits complex diffraction contrast at the

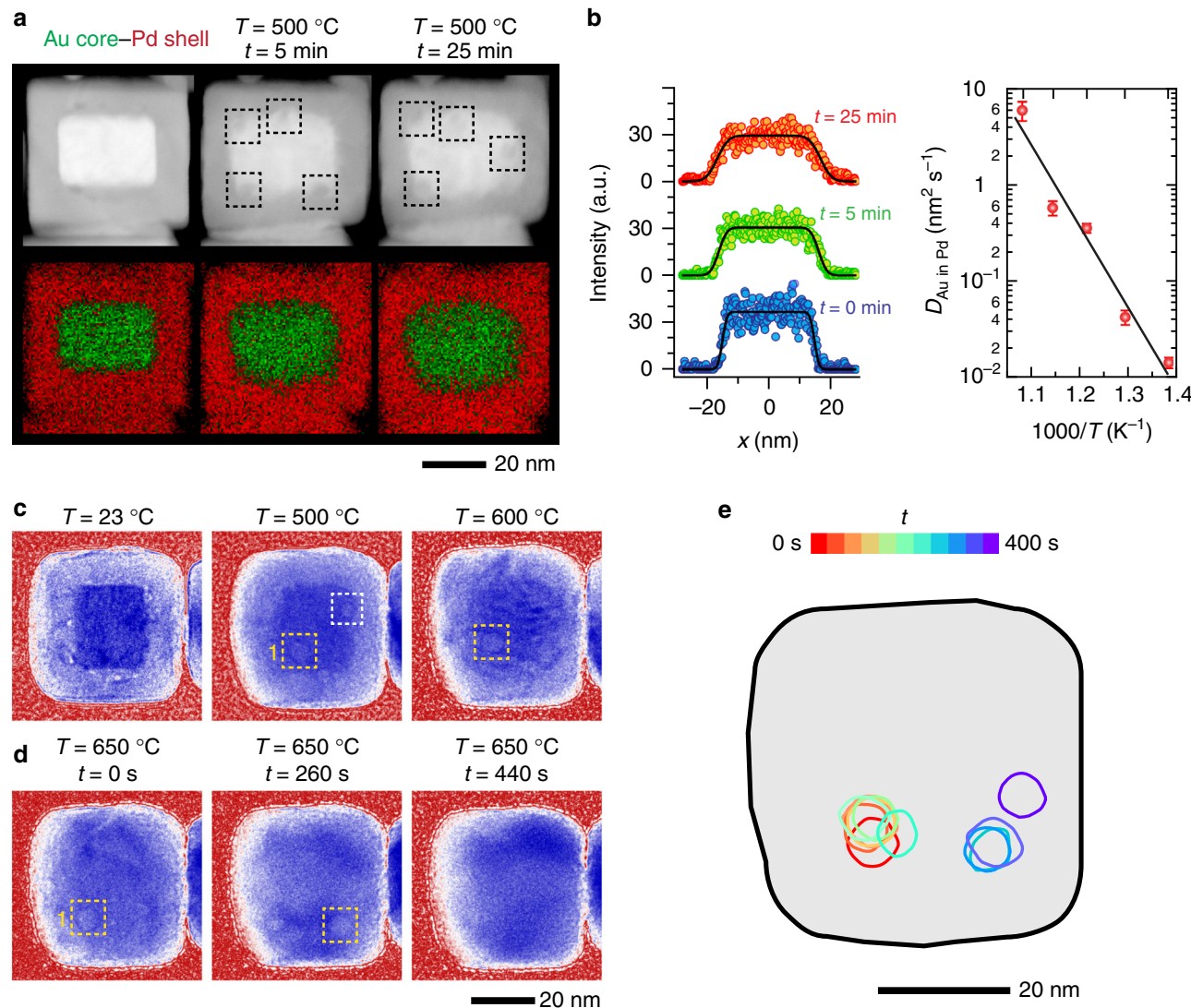

**Fig. 1** Heating-induced changes in the morphology of Au core–Pd shell (Au–Pd) nanoparticles (NPs). **a** In situ scanning transmission electron microscopy (STEM) images of the same NP after heating at 500 °C for 5 and 25 min, and their corresponding energy-dispersive X-ray (EDX) elemental maps. In the STEM images, voids show up as round areas of dark contrast. Dashed boxes highlight some of the voids that appeared in the NP during the heating. **b** (Left) Diffusion profiles extracted from the EDX measurements presented in the panel (**a**) and their respective error-function fits (black curves). (Right) Diffusion coefficients of Au diffusing in Pd obtained from extended studies (Supplementary Fig. 2), where we followed the diffusion of Au in these NPs as a function of temperature and time. The error bars indicate the standard deviation in the diffusion coefficients obtained at each temperature. An Arrhenius fit (black line) to the averaged diffusion coefficients gives the activation energy of $E_a = 1.6 \pm 0.2$ eV. **c** In situ transmission electron microscopy (TEM) image sequence of a NP during heating to 500 and 600 °C. In the TEM images, the voids (highlighted with dashed boxes) show up as dark rings with a light core. **d** Images recorded after increasing the temperature to 650 °C (Supplementary Movie 1). The time-stamps are pegged to time in the movie. The position of the void denoted as 1 is tracked over 400 s in **e**. At $t = 440$ s, the void annihilated at the surface on the right of the NP

beginning of the movie, but later, we can see how a dislocation interacted with a void. It is clear that the dislocation and void moved together. As the dislocation moved across the NP, the void was dragged to the surface, leading to the annihilation of the void and the formation of a residual notch at the NP surface. Hence, the intermittent void motion was caused by the random release of dislocations pinned by the void. In Fig. 1d, the dislocation driving the motion is less clear because the NP was not oriented in a way that established the right diffraction condition. Conversely, interesting interplays of dislocation and void motion can be captured in NPs under the right imaging conditions. Supplementary Movie 3 and Supplementary Fig. 5b show the image sequence captured from one such NP where three voids pin a single dislocation. In Supplementary Figs. 6 and 7, we further

describe the results from other NP geometries. Supplementary Fig. 6 shows an irregularly shaped NP that is found together with the cuboid NPs, whereas Supplementary Fig. 7 shows experiments where we used Au nanorods as the NP core[27]. These NPs also exhibited similar behavior of void formation and migration. More importantly, the migration of voids and the associated motion of dislocations captured in our results contradict common assumptions that voids are immobile and that they act as immobile obstacles to dislocation motion[28–31].

**Interdiffusion and void dynamics in Pd core–Au shell NPs.** Next, we reversed the NP geometry such that the faster-diffusing Pd is now the core and slower diffusing Au is the shell. In Fig. 2, we present images obtained from these Pd–Au NPs under

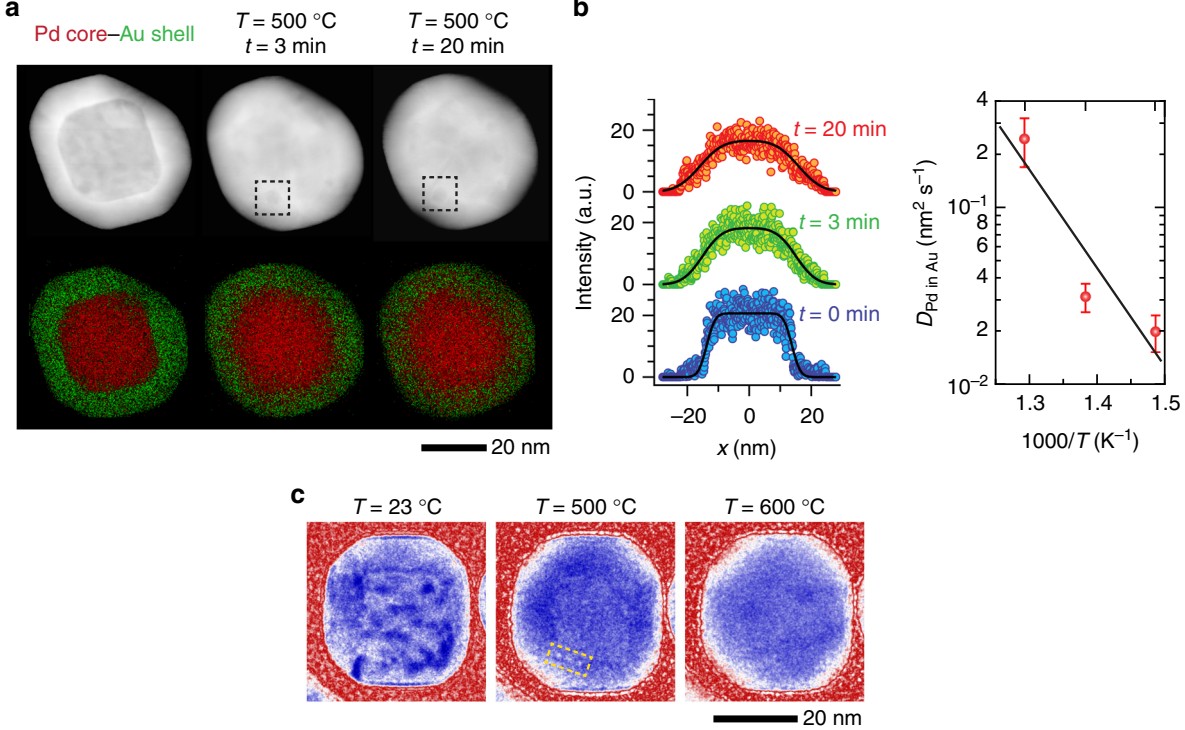

**Fig. 2** Heating-induced changes in the morphology of Pd core–Au shell (Au–Pd) NPs. **a** In situ STEM images of the same NP after heating at 500 °C for 3 and 20 min, and their corresponding EDX maps. The dashed box highlights a void in the NP. **b** (Left) Diffusion profiles extracted from the EDX measurements presented in the panel (**a**) and their respective error-function fits (black curves). (Right) Diffusion coefficients of Pd diffusing in Au obtained from extended studies (Supplementary Fig. 8), where we followed the diffusion of Pd in these NPs as a function of temperature and time. The error bars indicate the standard deviation in the diffusion coefficients obtained at each temperature. An Arrhenius fit (black line) to the averaged diffusion coefficients gives the activation energy of $E_a = 1.0 \pm 0.6$ eV. **c** In situ TEM image sequence of a NP during heating to 500 and 600 °C

experimental conditions identical to that described in Fig. 1. First, we observe that the bimetallic interface shifts inwards at 500 °C as expected (Fig. 2a). Further STEM-EDX experiments (Supplementary Fig. 8) indicate that the Pd–Au NPs become homogenous alloys within shorter heating times compared to Au–Pd NPs at the same temperature. It suggests a faster diffusion of Pd into Au, which is confirmed by a lower activation energy of $E_a = 1.0 \pm 0.6$ eV for the diffusion of Pd in Au. Compared to Au in Pd diffusion, this is a larger decrease from the reported lattice diffusion activation energy ($E_a = 1.8$ eV), but it is consistent with the drop reported for the grain-boundary diffusion activation energy ($E_a = 0.6$ eV)[21,22]. Second, nanoscale voids also formed in these NPs near the bimetallic interface but they were smaller than those in Au–Pd NPs. The small size of these voids makes it considerably harder to track the location and motion of these voids in the TEM images. Nevertheless, they did not lead to hollow core formation as we expected for NPs with a faster diffusing core. Similar to the behavior observed in Au–Pd NPs, voids migrated to the surface and were annihilated.

In short, both NP systems behave differently from the models based on conventional KE described in refs. [23,24], where binary core–shell NPs with a faster diffusing core (i.e., Pd) are expected to transform into hollow core NPs. The discrepancy between our observations and the models may arise because these conventional models only considered the atomic and vacancy fluxes. Au and Pd have a lattice mismatch of ~5% (4.08 and 3.89 Å, respectively[20]), and so, there can be a significant residual strain. Electron diffraction and high-resolution STEM images confirm that our NPs have a coherent bimetallic interface (Supplementary Figs. 9 and 10), which means that the interface is likely strained.

**Vacancy formation energies from density functional theory (DFT).** To identify the correct mechanism behind the observed behavior, we performed DFT calculations and compared the vacancy formation energies in the metallic lattice site and at the Au–Pd interface. The formation energies were calculated for two configurations: one where the Au lattice is fixed and the Pd lattice can relax (Au–Pd NPs), and another where the Pd lattice is fixed while the Au lattice can relax (Pd–Au NPs) to simulate the effect of strain. In both cases, the calculations (Supplementary Table 1) show that Au vacancies formed at Au lattice sites are more stable than those formed at the interfacial sites. Hence, the Au vacancies will not diffuse into the Pd, suppressing the expected vacancy flux into Pd due to KE, which prevents the formation of a hollow core. On the other hand, Pd vacancies are more stable at the interface compared to Pd lattice sites. Our calculations indicate that the vacancies formed in Pd will further diffuse into Au as the migration is energetically downhill. As shown in Fig. 3a, the formation energy decreases for both simulated configurations if the vacancy migrates across the interface and into the Au bulk lattice. For Au–Pd NPs, the vacancy flux is directed towards the Au core. Hence, the center of the NP becomes saturated with vacancies. The lack of structural defects in the core further means that the accumulated vacancies can only condense at interfacial defects that were originally present in the NPs, such as stacking faults at the bimetallic interface[32–35]. For Pd–Au NPs, the vacancy flux is directed towards the Au shell. Here, the vacancies are free to diffuse outwards towards the NP surface. Subsequent annihilation of vacancies at the Au surface will reduce the number of vacancies available for void formation, leading to the smaller voids found in this system (Fig. 2c). The decrease in activation energy for Pd

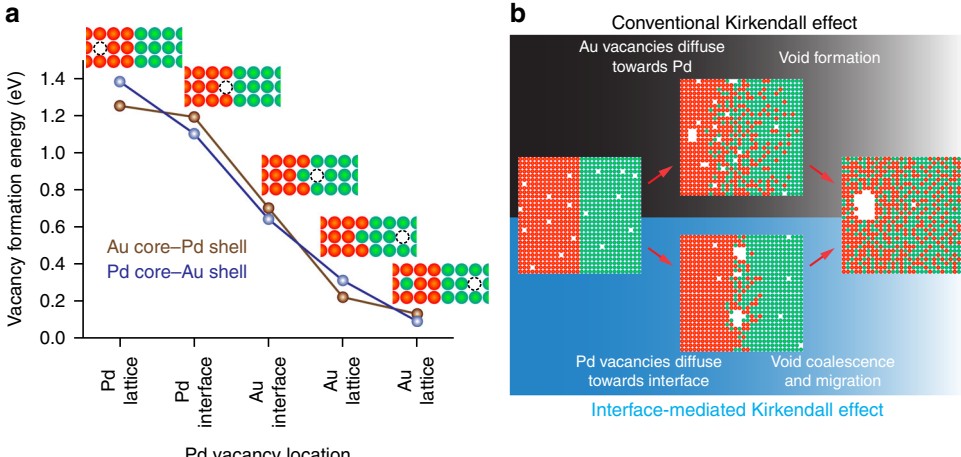

**Fig. 3** Formation energy of Pd vacancies and a schematic illustrating interface-mediated Kirkendall effect (KE). **a** Density functional theory (DFT) calculations of the formation energy of a Pd vacancy at different locations of the Au–Pd layered structure (Supplementary Fig. 11). Migration of the Pd vacancy across the Au–Pd interface is accompanied by diffusion of a Au atom into a Pd lattice site. **b** A schematic comparison of the conventional KE and the observations of interfacial void formation and void migration in this study. The conventional view for KE is that interdiffusion leads to a flux of vacancies towards the faster-diffusing side. When the vacancy concentration in the diffusion zone reaches supersaturation, the vacancies can condense at defect sinks on the faster-diffusing side to form voids that grow with time[12,36]. In our study, we see void formation at the bimetallic interface due to the clustering of vacancies at the interface leading to void formation and subsequent void migration to the surface. Green and red spheres represent Au and Pd atoms, respectively

diffusing into Au may also be explained by this flow of vacancies. We also highlight here that the intermixing of Au and Pd (Au atom diffusing into the Pd lattice) leads to a further drop in the vacancy formation energy, which is consistent with observed alloying of the two metals. Hence, our results can be described as a novel manifestation of KE, where the inherently higher vacancy formation energy in Pd leads to a vacancy flux towards Au, the slower diffusing species. This situation is opposite from the conventional KE where the vacancy flux is towards the faster diffusing species (Fig. 3b).

**Void dynamics in sandwich Au–Pd–Au shell NPs**. To test the hypothesis that interfaces are indeed the critical sites for void formation, we deposited another layer of Au on top of the Au–Pd NPs, creating a Au–Pd–Au sandwich structure. As shown in Fig. 4a, b, voids are seen at both bimetallic interfaces at 500 °C. The voids in these Au–Pd–Au NPs are larger, ~10 nm in diameter versus ~5 nm in diameter in the Au–Pd NPs. This increase in void size is explained by the additional Au layer eliminating the free Pd surface and trapping the vacancies at the outer Au–Pd interface. More significantly, these larger voids also migrated from the center of the NP towards the surface (Fig. 4b) when the NPs were further heated to 600 and 650 °C (Supplementary Movies 4 and 5), providing more evidence that the voids are mobile.

These observations of void migration and annihilation at the surface contradict theoretical models where nanoscale voids in binary alloy NPs are expected to shrink over time via the emission of vacancies[23,24,37]. The key premise in these models is that hollow NPs should be thermodynamically unstable due to the Gibbs–Thomson effect and the void will eventually shrink due to the vacancies diffusing from the inner surface towards the outer surface[38]. Void migration is a phenomena more commonly associated with the large thermal gradients in bulk materials, such as fusion materials[39] and with electromigration in microelectronic interconnect structures[40]. The absence of void shrinkage through vacancy emission is not an anomaly due to the higher vacancy formation energy of Pd because the voids migrated towards the surface in all three NP geometries, two of which have

Au shells. Therefore, our results suggest that void migration is instead the general mechanism for void removal from NPs.

In conclusion, we showed that nanoscale voids in bimetallic Au–Pd NPs prefer to form at the bimetallic interface, rather than at the core inside the NPs as predicted by theories based on the conventional picture of KE. It is not due to a discrepancy with theory. Rather, our results illustrate the effects of material properties (i.e., vacancy formation energies and NP structure) have on moderating the expected vacancy fluxes. Our DFT calculations suggest that the observed behavior is driven by a change in the direction of vacancy diffusion due to the higher vacancy formation energy of Pd compared to Au and is not a result of interfacial strain between the two metals. Hence, we anticipate interfacial voids to be a common intermediate structure in any bimetallic system with the tendency to alloy but has a significant difference in vacancy formation energies. In this case, the formation of nanoscale voids will occur together with both alloying and de-alloying processes and careful manipulation of this process may lead to new ways of engineering bimetallic NPs. These results also agree with voids being a kinetically stabilized state in NPs. However, our ability to track nanoscale changes during materials transformations demonstrate that voids neither shrink nor grow through the emission and diffusion of vacancies. Instead, nanoscale voids migrate through the near-perfect nanocrystals towards the surface. In addition, the correlated motion of a void and a dislocation suggests that nanoscale voids may not contribute as expected to the hardening of materials in general, where voids are assumed to be immobile obstacles to dislocation motion. These insights have broad implications for our understanding of how voids grow in bulk metals and of how nanoscale voids impact the mechanical properties of these metals.

## Methods
**Materials**. Gold (III) chloride trihydrate (HAuCl$_4$·3H$_2$O, Cat. No. 520918, Sigma-Aldrich Co., St. Louis, MO, USA), sodium borohydride (NaBH$_4$, Cat. No. 213462, Sigma-Aldrich Co., St. Louis, MO, USA), copper sulfate (CuSO$_4$, Cat. No. 451657, Sigma-Aldrich Co., St. Louis, MO, USA), L-ascorbic acid (C$_6$H$_8$O$_6$, Cat. No. A5960-25G, Sigma-Aldrich Co., St. Louis, MO, USA), cetyltrimethylammonium bromide (CTAB, Cat. No. 52370-500G, Sigma-Aldrich Co., St. Louis, MO, USA), palladium (II) chloride (PdCl$_2$, Cat. No. 323373-1G, Sigma-Aldrich Co., St. Louis, MO, USA),

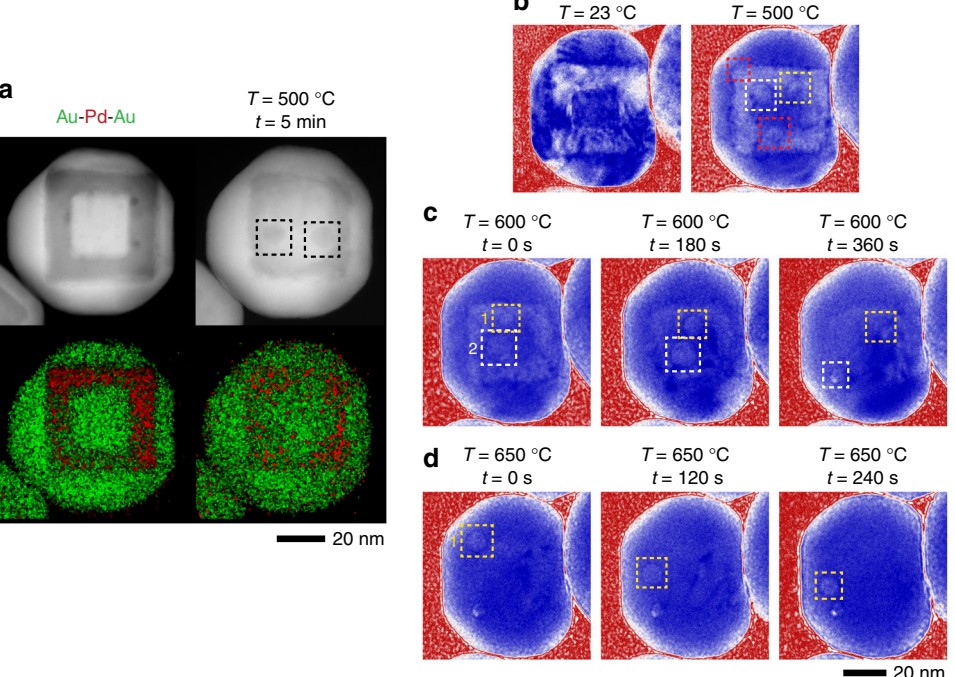

**Fig. 4** Void formation and migration in Au–Pd–Au sandwich NPs at elevated temperatures. **a** In situ STEM (upper) and EDX (lower) images of the as-synthesized NP morphology and after heating to 500 °C. The images indicate that the voids formed in these NPs are larger than the bi-layer NPs. Voids are denoted by dashed squares. **b–d** In situ TEM image sequences of void migration in a Au–Pd–Au NP after heating to 500 °C, 600 °C (Supplementary Movie 4), and further to 650 °C (Supplementary Movie 5). The time-stamps in **c** and **d** are pegged to time in the supplementary movies. Note that void 1 increased in size between $t = 0$ s and $t = 180$ s at 600 °C. It can be seen from Supplementary Movie 4 that this increase was due to two void coalescence events between $t = 60$–80 s. Void 2 got annihilated at the surface between $t = 180$ s and $t = 360$ s at 600 °C (Supplementary Movie 4), resulting in a small hole on the NP surface. The sequence at 650 °C shows the extended migration of void 1 (Supplementary Movie 5). Movie 5 is recorded 70 s after the end of Movie 4

hydrochloric acid (HCl, Cat. No. 84415-500ML, Sigma-Aldrich Co., St. Louis, MO, USA), and sodium tetrachloropalladate (II) (Na₂PdCl₄, Cat. No. 205818-1G, Sigma-Aldrich Co., St. Louis, MO, USA) were used as received without further purification. All aqueous solutions were prepared using deionized water with a resistivity of 18.2 MΩ cm.

**Synthesis of Au core–Pd shell NPs.** We synthesized the Au nanocube cores following the protocol described by Sun et al. [41] A Au seed solution was prepared by adding 150 μL of 100 mM ice-cold aqueous NaBH₄ solution into a mixture of 6.25 mL of 1 mM HAuCl₄·3H₂O and 18.75 mL of 0.1 M aqueous CTAB solution. Then, this seed solution was left undisturbed for 4 h.

To grow the Au nanocubes, 20 mL of 20 mM CTAB was added into a clean polypropylene tube, followed by the addition of 5 mL of 2 mM HAuCl₄·3H₂O aqueous solution. Then, we added 50 μL of 10 mM CuSO₄ aqueous solution, 3 mL of 100 mM ascorbic acid, and 5 μL of the Au seed solutions sequentially. The solution was gently mixed by inversion of the test tube after the addition of each component. Next, we transferred 1 mL of this final solution into a 1.5 mL centrifuge tube. Centrifugation and redispersion in deionized water were performed twice at 10,000 rpm for 5 min. Finally, the Au nanocubes were dispersed in 1.8 mL of deionized water and transferred into a clean glass vial.

To deposit the Pd shell, 200 μL of 100 mM CTAB, 60 μL of 50 mM of ascorbic acid, 300 μL of 10 mM Na₂PdCl₄ were added into the above Au nanocube solution sequentially, and the mixture was heated at 80 °C for 10 min in a water bath. After cooling down to room temperature, the mixture was washed twice at 10,000 rpm for 5 min. The final NPs were dispersed in 1 mL of deionized water.

**Synthesis of Pd core–Au shell NPs.** We synthesized the Pd nanocubes following the protocol described by Niu et al. [42] Firstly, the 10 mM H₂PdCl₄ solution was prepared by dissolving 0.173 g of PdCl₂ in 10 mL of 200 mM HCl. Next, we prepared the Pd seed solution by adding 1 mL of 10 mM H₂PdCl₄ into 20 mL of 12.5 mM aqueous solution of CTAB, and the mixture was heated at 95 °C for 5 min under stirring. Then, we added 160 μL of 100 mM ascorbic acid into the mixture and waited for 20 min, after which the Pd seed solution was allowed to cool down to room temperature.

To synthesize the Pd nanocubes, 5 mL of 100 mM aqueous CTAB solution, 125 μL of 10 mM H₂PdCl₄ was mixed with 40 μL of the seed solution and 50 μL of 100 mM ascorbic acid and kept at 60 °C for 2 h in a water bath. After cooling down to room temperature, we transferred 1 mL of the Pd nanocube solution into a 1.5 mL centrifuge tube. Centrifugation and redispersion in deionized water were performed twice at 10,000 rpm for 10 min, and the Pd nanocubes were dispersed in 200 μL of deionized water. Next, we mixed 200 μL of this as-prepared solution with 200 μL of 0.1 M CTAB, 60 μL of 50 mM ascorbic acid, 300 μL of 10 mM HAuCl₄, and the mixture was heated at 80 °C for 10 min in a water bath. After cooling down to room temperature, the mixture was washed twice by centrifugation at 10,000 rpm for 5 min and redispersion. Finally, the NPs were dispersed in 200 μL of deionized water.

**Synthesis of Au core–Pd shell–Au shell NPs.** We mixed 100 μL of the as-prepared Au core–Pd shell NPs solution (see above) with 100 μL of 0.1 M CTAB, 30 μL of 50 mM ascorbic acid, 150 μL of 10 mM HAuCl₄ sequentially, and this solution was heated at 80 °C for 10 min in a water bath. After cooling down to room temperature, the mixture was washed twice at 10,000 rpm for 5 min. Finally, the Au–Pd–Au NPs were dispersed in 100 μL of deionized water.

Further details about the as-synthesized NPs can be found in the supplementary notes.

**TEM imaging and EDX spectrometry.** Three TEMs were used for this study. The STEM and EDX analysis were performed on a 300 kV Titan TEM (Thermo Fisher Scientific, Hillsboro, OR, USA) with a Bruker XFlash 6TI30 EDX spectrometer (Bruker Nano Analytics, Berlin, Germany) and a probe corrected 200 kV JEOL ARM200CF (JEOL Ltd., Tokyo, Japan) with an Oxford X-MaxᴺN 100TLE SDD EDX spectrometer (Oxford Instruments, Abington, Oxfordshire, UK). The in situ heating experiments were conducted on a 200 kV JEOL-2010F field emission transmission electron microscope (JEOL Ltd., Tokyo, Japan) equipped with a Gatan OneView-IS camera (Gatan Inc., Pleasanton, CA, USA).

The samples were heated using Wildfire heating holders from DENSsolutions (DENSsolutions, Delft, Netherlands). The NP solution was first drop-casted onto the heating chips supplied by DENSsolutions and then allowed to dry. We mention here that these NP samples were not plasma cleaned, a common procedure used to prepare samples for TEM characterization. Plasma cleaned samples changed their surface morphology, even when only heated to relatively mild temperatures for short durations (Supplementary Fig. 12). We also found that performing STEM-EDX mapping at elevated temperatures lead to a restructuring of the NPs that was not found in NPs heated in the absence of the electron beam.

Hence, we acquired the STEM images and EDX maps by heating the NPs for fixed durations with the electron beam blanked and then, only characterized the samples after they had cooled back to ambient temperature to reduce the influence of the electron beam on NP morphology. For the same reason, the in situ TEM image sequences were acquired with a relatively low electron flux of 60–70 e−/ (Å2 s). The images were extracted from time points that are a few seconds after the heater had reached the designed temperature to allow for thermal stabilization.

**DFT calculations**. First-principles DFT calculations are implemented via the Perdew–Burke–Ernzerhof (PBE) exchange-correlation based on the generalized gradient approximation (GGA)[43,44] in the Vienna ab initio Simulation Package (VASP)[45,46]. The projector augmented-wave (PAW) method[47] was used to describe the electron–ion interactions. Plane-wave cutoffs were set to 520 eV. As the NPs are fairly large, we modeled the Au–Pd interfaces using surface models with supercell structures along the fcc (001) directions comprising six layers of Au and six layers of Pd, in which the middle two layers of Au or Pd were treated as bulk-like (Supplementary Fig. 11). To represent a Au–Pd (Pd–Au) NP, the overall supercell parameters followed that of the Au (Pd) bulk lattice parameters. Structural optimizations were carried out by relaxing the atomic coordinates of all atoms until the calculated Hellmann–Feynman forces on each atom were <0.01 eV/Å, except for atoms in the 'bulk' layer (the third layer from the interface) that were kept to their bulk lattice parameters. The Brillouin zone was sampled by adopting $3 \times 3 \times 1$ Monkhorst–Pack (MP) $k$-point meshes[48].

We compute the vacancy formation energies $\Delta E$ using the following expression adapted from ref. [49]:

$$\Delta E = E_{vac} - E_{core-shell} - E_{bulk} \quad (1)$$

where $E_{vac}$ and $E_{core-shell}$ are the total energies of the interface supercell containing one and no vacancy, respectively, while $E_{bulk}$ is the energy per atom of the corresponding bulk structure (of the removed element). For each of the core–shell NP type, we considered different vacancy types, i.e., Au and Pd vacancies in both NP configurations, Au–Pd and Pd–Au. Their $\Delta E$ values as a function of their location in the model are tabulated in Supplementary Table 1. For Au vacancies, the lowest $\Delta E$ occurs in the bulk Au. For Pd vacancies, $\Delta E$ at the interface is lower than that of bulk Pd, and so, we further explored if the vacancy would continue the migration across the interface and into the Au layer. Here, we performed simulations where (1) the Pd vacancy migrates across the interface leading to a Au atom at the interface moving in to occupy the lattice position of the Pd vacancy, and (2) migration of the vacancy deeper into the Au lattice.

## Data availability
The data that support the findings of this study are available from the corresponding author upon reasonable request.

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

## Acknowledgements

This work was supported by the Singapore National Research Foundation's Competitive Research Program funding (NRF-CRP16-2015-05).

## Author contributions

S.W.C. and U.M. conceived and designed the experiments. S.W.C. performed the electron microscopy characterization and in situ heating experiments. Z.M.W. and T.L.T. performed the DFT calculations. Z.B. performed the image processing and the data analysis of the EDX maps. S.F.T. synthesized the NPs. All authors contributed to the discussion and writing of the manuscript.

## Additional information

**Competing interests:** The authors declare no competing interests.

