## [Peer Review File · Nature Communications]

Reviewers' Comments:

Reviewer #1:

Remarks to the Author:

This study reports a discovery of novel void diffusion mechanisms in bimetallic nanoparticles (NPs). The findings are important as they show how material properties can alter and constrain expected void migration mechanisms based on the Kirkendall Effect (KE) observed in many other bimetallic systems. This work is of great scientific interest to the nanoparticle, nano-metallic materials, and electron microscopy research communities.

This communication uses sound judgement and presents original results. The authors use temporal and spatially resolved in-situ transmission electron microscopy to track void formation and diffusion in Au/Pd NPs. By annealing two different Au/Pd nanoparticle designs - i.e. Au core with Pd shell vs. Pd core and Au shell - voids were tracked at various time steps and shown to migrate to the NP surface and annihilate in both cases. Conventional KE predicts a hollow-core NP for the Pd core and Au shell geometry as Pd is the faster diffuser and would diffuse outward as vacancies diffuse into the core. Instead, the experiments revealed void formation at the Au-Pd interface that diffuse outward and in the same direction as Pd atoms. This reversal in void diffusion was also shown in the Au core, Pd shell geometry. This is clearly and effectively shown in the supplementary movies.

The authors' claim that this is not a contradiction to KE theory, but instead a new manifestation of KE is plausible. It is shown in STEM that, because voids form at the bimetallic interface, the expected vacancy flux can indeed be moderated when annihilation of vacancies at the Au surface reduces the number of vacancies available for void formation. This conclusion is also successfully supported by DFT calculations.

The manuscript is exempt of major grammatical errors, faulty judgements, or misleading interpretations.

In summary, the text is easy to follow and exceptionally written. The figures and illustrations are well presented, complete, and accurate.

Reviewer #2:

Remarks to the Author:

The paper claims that unlike monometallic nanoparticles that evolve in hollow core NPs, in bimetallic core-shell NPs (here Au-Pd) nanoscale voids occur at the bimetallic interface and eliminate at the surface during heat treatment via migration without requiring a vacancy emission mechanism. This claim is supported by STEM and TEM observation of NP morphologies that lead to analyze the role of nanoscale diffusion fluxes and DFT calculations of the formation energy of vacancies of both sides of the interface.

The paper, with its supplementary information, is convincing and deserves publication as it proposes a possible new scenario for void annihilation in bimetallic NPs. Nevertheless, some piece of quantitative information are lacking and stops the reader to admit a general reproducibility of the mechanism. These complements are considered as minor nonetheless mandatory revisions, as they would benefit a large audience.

1. Please indicate the characteristics of the NP distribution : size, mean size, etc... And possible variation in configurations. This would lead to evaluating the reproductibility of the mechnaism and

should result in qualifying if one can consider the NP as « isolated » or if other neighbouring particles could serve as a reservoir for possible matter exchange. This is not clear from Figure 1 whether voids could transfer at the interface between 2 NPs.

2. The void-dislocation interaction is definitely something worth digging. Can you be more explicit about this behavior ? Do you have enough statistics or is this just a rare (unique ?) event ?

3. Could you comment on the possible role of the NP shape regarding the void annihilation on the surface ? Can it be expected to be extended to noncubic NP?

4. Could you conclude on which couple of metals would lead to the same behavior of NPs ? for instance, you could expect the same scenario for large size-mismatch particles? Isn't the small size mismatch a necessary driving force in your underlying assumption of quasi-coherent interfaces ?

Reviewers' comments:

Reviewer #1 (Remarks to the Author):

This study reports a discovery of novel void diffusion mechanisms in bimetallic nanoparticles (NPs). The findings are important as they show how material properties can alter and constrain expected void migration mechanisms based on the Kirkendall Effect (KE) observed in many other bimetallic systems. This work is of great scientific interest to the nanoparticle, nano-metallic materials, and electron microscopy research communities.

This communication uses sound judgement and presents original results. The authors use temporal and spatially resolved in-situ transmission electron microscopy to track void formation and diffusion in Au/Pd NPs. By annealing two different Au/Pd nanoparticle designs - i.e. Au core with Pd shell vs. Pd core and Au shell - voids were tracked at various time steps and shown to migrate to the NP surface and annihilate in both cases. Conventional KE predicts a hollow-core NP for the Pd core and Au shell geometry as Pd is the faster diffuser and would diffuse outward as vacancies diffuse into the core. Instead, the experiments revealed void formation at the Au-Pd interface that diffuse outward and in the same direction as Pd atoms. This reversal in void diffusion was also shown in the Au core, Pd shell geometry. This is clearly and effectively shown in the supplementary movies.

The authors' claim that this is not a contradiction to KE theory, but instead a new manifestation of KE is plausible. It is shown in STEM that, because voids form at the bimetallic interface, the expected vacancy flux can indeed be moderated when annihilation of vacancies at the Au surface reduces the number of vacancies available for void formation. This conclusion is also successfully supported by DFT calculations.

The manuscript is exempt of major grammatical errors, faulty judgements, or misleading interpretations.

In summary, the text is easy to follow and exceptionally written. The figures and illustrations are well presented, complete, and accurate.

We are grateful for the reviewer's generous remarks about the manuscript and enthusiasm for the research.

Reviewer #2 (Remarks to the Author):

The paper claims that unlike monometallic nanoparticles that evolve in hollow core NPs, in bimetallic core-shell NPs (here Au-Pd) nanoscale voids occur at the bimetallic interface and eliminate at the surface during heat treatment via migration without requiring a vacancy emission mechanism. This claim is supported by STEM and TEM observation of NP morphologies that lead to analyze the role of nanoscale diffusion fluxes and DFT calculations of the formation energy of vacancies of both sides of the interface.

The paper, with its supplementary information, is convincing and deserves publication as it proposes a possible new scenario for void annihilation in bimetallic NPs. Nevertheless, some piece of quantitative information are lacking and stops the reader to admit a general reproducibility of the mechanism. These complements are considered as minor nonetheless mandatory revisions, as they would benefit a large audience.

We are grateful for the reviewer's helpful comments about improving the manuscript and addressed the comments accordingly below.

1. Please indicate the characteristics of the NP distribution : size, mean size, etc... And possible variation in configurations. This would lead to evaluating the reproductibility of the mechnaism and should result in qualifying if one can consider the NP as « isolated » or if other neighbouring particles could serve as a reservoir for possible matter exchange. This is not clear from Figure 1 whether voids could transfer at the interface between 2 NPs.

We have added details about the nanoparticles (NPs) to the supplementary information (SI). The average sizes of the NP cores and core-shell NPs have been added as the first section of the SI and low magnification of images of the drop-casted NPs have been added as Supplementary Figure 1.

Although the NPs prefer to cluster, it is possible to find isolated NPs or NPs with limited contact with other NPs for the STEM-EDX analysis. We also believe that there is minimal mass transport between the NPs due to the stabilizing effect of residual cetyltrimethylammonium bromide (CTAB.) One of us (SF Tan) has previously shown that molecular coatings can stabilize NP morphology during STEM imaging (Langmuir 2017, 33, 1189-1196), even though CTAB was not one of the molecules studied.

We have not observed instances of void transfer at the NP-NP interface.

To address the reviewer's concern about matter exchange, we now include an additional experiment as Supplementary Figure 12. Here, the Pd-Au NPs that had been plasma cleaned, a common surface preparation procedure for TEM samples, are imaged after 1 min of heating at 400 °C. It is clear that these NPs are unstable even at such a mild heating temperature and exhibit significant agglomeration, which supports the stabilizing effect of CTAB.

These observations suggest that external mass transport during heating is not a significant concern with our CTAB stabilized NPs. To ensure that the readers of Nature Communications can reproduce these experiments, we have added a note about not plasma-cleaning the NPs to the methods.

2. The void-dislocation interaction is definitely something worth digging. Can you be more explicit about this behavior? Do you have enough statistics or is this just a rare (unique?) event?

The void-dislocation interaction is not a rare event. We have seen multiple instances of this void-dislocation interaction and we believe it is general across all the NPs. The difficulty in observing these events is that we do not know beforehand which NPs are oriented in the right diffraction conditions for the dislocations to show up with strong contrast.

To address the reviewer's comment, we added another image sequence as Supplementary Figure 5(b) and Supplementary Movie 3. This sequence is very interesting, because we can see the dynamics of three voids that appeared to be pinned by a single dislocation. We also added a few sentences in page 5 to describe this sequence.

3. Could you comment on the possible role of the NP shape regarding the void annihilation on the surface? Can it be expected to be extended to noncubic NP?

As shown in Supplementary Figure 1, there is usually some NPs that are irregularly shaped mixed in together with the cuboid shaped ones. We have also seen void annihilation in these NPs and so, we expect the behaviour to be general. The image sequence of one such NP is added as supplementary Figure 6(a).

4. Could you conclude on which couple of metals would lead to the same behavior of NPs? for instance, you could expect the same scenario for large size mismatch particles? Isn't the small size mismatch a necessary driving force in your underlying assumption of quasi-coherent interfaces?

According to the DFT calculations, the driving force comes from the large difference in vacancy formation energy and not lattice strain. Voids should still be present in cases where the interface is less coherent. Figure 3 also shows that chemical intermixing energy (from the diffusion of the Au atom into the Pd atom) produces a more pronounced drop in vacancy formation than a change in simulated lattice strain, which also confirms that the two metals tend to alloy. Hence, we do not expect lattice mismatch in the bimetallic NPs and a quasi-coherent interface to be key requirements for our reported behaviour to be observed. Conversely, the metal couples that alloy and have a significant difference in vacancy formation energy should show the same behaviour.

To test this hypothesis, we added experiments where we used Au nanorods as the core of the core-shell NPs as Supplementary Figure 7. Here, we expect the interface coherence to be poorer compared to the cuboid cores because of the round tips and curved surfaces of the nanorods. We see the same behaviour of void formation and migration observed in these NP.

We now highlight in Page 7, paragraph on that the DFT being consistent with the metals alloying.

“We also highlight here that the intermixing of Au and Pd (Au atom diffusing into the Pd lattice) leads a further drop in formation energy, which is consistent with observed alloying of the two metals.”

We have revised the conclusion to emphasize the results from the DFT calculations (Page 9, paragraph 2).

“Our DFT calculations suggest that the observed behavior is driven by a change in the direction of vacancy diffusion due to the higher vacancy formation energy of Pd compared to Au, and is not a result of interfacial strain between the two metals. Hence, we anticipate interfacial voids to be a common intermediate structure in any bimetallic system with the tendency to alloy and has a significant difference in vacancy formation energies.”

Reviewers' Comments:

Reviewer #2:

Remarks to the Author:

The second version of the submitted manuscript provides a clear guidance of many mandatory milestones of the authors' study; the added material is clear, qualitative and quantitative.

I recommend publication without corrections